# Transport Task Models with Variable Supplier Availabilities

**Julian Vasilev** [1,*] , **Rosen Nikolaev** [2] and **Tanka Milkova** [2,*]

1    Department of Informatics, University of Economics Varna, 9002 Varna, Bulgaria
2    Department of Statistics and Applied Mathematics, University of Economics Varna, 9002 Varna, Bulgaria; nikolaev_rosen@ue-varna.bg
*    Correspondence: vasilev@ue-varna.bg (J.V.); tankamilkova@ue-varna.bg (T.M.)

**Abstract:** *Background*: With regard to the definition of an optimal transport plan for some material flow in the logistics system in literature and practice, the classical transport task model is developed. The minimization of total transport costs is usually considered for optimality. Some modifications to the classical transport task have also been developed. *Methods*: The article uses the methods of linear optimization. Based on these methods, two modified transport task models have been constructed, which consider the possibility of planning in advance the quantities available from the suppliers of the transported cargo. These models are applicable in SCM for pharmaceuticals with a national logistics hub. Furthermore, a solver in MS Excel is used to determine the optimal solution of optimization models. *Results*: Two new (modified, extended) models of the transport task have been constructed, in which a preliminary planning of the available quantities of the transported cargo at the suppliers is made. These quantities shall be planned in such a way as to ensure a minimum total transport cost. *Conclusions*: By applying the proposed new transport task models, lower total transport costs for carrying out imported pharmaceuticals can be ensured compared to an application of the classical transport task model.

**Keywords:** optimization; transportation problems; logistics system; national logistics hub

## 1. Introduction

One of the main tasks of any rationally functioning business organization is related to the need to make efforts to preserve its competitive advantages and increase the efficiency of its operation. In order to achieve these goals, it is necessary to seek and apply different opportunities for the optimal implementation of all activities. One major activity associated with significant financial costs is the need for the physical movement of material flows. This necessitates the application of various science-based methods and approaches to determine optimal management decisions.

A number of methods and approaches are available in the specialized literature, leading to optimal solutions regarding the transport of material flows [1], but the specific features of each specific economic system necessitate the continuous adaptation and further development of existing models and methods of optimization. One of the classic methods related to providing opportunities for optimizing transport activities is the famous model of transport tasks. Numerous modifications of this model are also known in the scientific literature.

Nowadays is the post-COVID-19 period [2,3]. Many supply chains are modified. Many air carriers transformed their passenger airplanes into cargo ones. Late supplies from Eastern and Central Asia to Europe are often practiced in many industries. Theories concerning Vendor management inventories (VMI), stock replenishment, and supply chain management (SCM) helped the logistics managers find adequate solutions to the recent changes in supply chains. These theories and good logistics practices were useful before and during COVID-19, but nowadays the search for new theories and good practices for inventory movement (and transportation) and inventory replenishment is needed. That is

why the authors of this article, on the basis of their professional knowledge in mathematics (especially in the sphere of the transport task models), from one side, and their direct observations in the sphere of retail with the pharmaceutical industry, from another side, have created two new models for the transport task. The novelty of these models is that they are not only theoretical but also practical for other retail businesses. The novelty of the proposed model gives the possibility of extending existing transportation models and making them adequate to the up-to-date reality of SCM in pharmaceuticals.

The structure of the paper is according to the well-known IMRAD format. The introduction outlines the current state of supply chains and the need for new transportation models. The literature review discusses the current state of the transport model and its extensions. The section "Methods and Research Designs" gives a formal description of the created models. The "Results" section presents the results and shows some comparisons of the performance of the created models compared to the classical transport task model. The conclusion section summarizes the findings, outlines future research, and gives practical implications.

## 2. Literature Review

The most well-known is the classical model of the transport task [4–6], the purpose of which is to determine an optimal plan of transport at certain starting points with known quantities of stocks in them and certain reception points with known quantities of demand. As a criterion for optimality in the transport task, minimization of total transport costs is usually considered [7–9], i.e., at certain transport costs from all points of departure [10,11] to all reception points, it is necessary to define a transport plan [12] that satisfies the stated needs of the receiving points with the available quantities of stocks at the points of departure at minimum total transport costs [13–15]. This model is known as the transport problem by criterion value [16–18]. Some modifications of the transport task by criterion value are also well known [7,19], taking into account the specifics of transport activities. For example, when it is impossible to carry out direct carriage of goods between a specific point of departure and a specific receiving point, it is transferred to the so-called Freight Point—transport task with blocked transports. Another example of a modification of the transport task by criterion value is the two-stage transport task, which is passed in cases where it is necessary for transported goods to pass through intermediate warehouses before reaching the point of departure to the receiving point.

In some cases, the nature of the goods carried and the operations carried out with them could require the distribution of cargo between points of departure and reception to take place as soon as possible. In such cases, it is necessary, in addition to the points of departure with their available quantities and the points of destination with their stated needs, to allow time for moving the cargo from each point of departure to each receiving point. Furthermore, the optimality criterion "minimum total transport costs" is replaced by the optimality criterion "minimum transport execution time", and the model [20] thus obtained is called the transport problem by time criterion. The time taken for the completion of the entire plan of carriage shall be taken as the greatest time of movement from a point of departure to a point of destination where carriage is to be carried out. To find the optimal solutions to the problems thus set, well-known methods of linear optimization are used.

Solving a transport task considering only one factor—transport costs or transport execution time—is not always justified [21]. Modification of the transport task with simultaneous consideration of the time and cost of plan implementation are known.

A major limitation of the classical transport task is that a homogeneous load (one type of cargo) is carried. In view of this, modifications to a multiproduct transport task that provides minimum total transport costs subject to certain restrictive conditions in the process of delivering several different types of products from one supplier to multiple users are also proposed in the literature [22–24].

In all modifications of a transport task (by criterion, total costs) that are described, the availability of the suppliers and the needs of the users [25] have been determined

in advance. On the basis of these stocks and stated needs, an optimal transport plan is determined, which ensures a minimum total cost of transportation.

The previous studies do not cover modified versions of the classical transportation task where the SCM is with a national logistics hub (serving as a delivery point for imported pharmaceuticals), several suppliers, and several corporate customers, where the quantities in the suppliers may vary, and where additional restrictions to the quantities in the suppliers are present.

In this paper, the authors aim to present a modification of a transport task in which preplanning of the suppliers' availabilities is carried out. New constraint conditions are introduced that provide inventory planning so as to obtain smaller total transport costs compared to the classical transport task model [26–29]. The constructed transport task models can be easily solved using suitable, specially developed software products [30–33]. Furthermore, corporate KPIs may be improved [34–37].

## 3. Methods and Research Design

### 3.1. Existing Transport Task Models

The classical model of a transport problem is used as a basis for creating the new models. The formal description of the classical model of a transport problem is, in general, as follows:

$m$, at starting points $A_1, A_2, \ldots, A_m$ (called suppliers), has some homogeneous products, respectively in quantities $a_1, a_2, \ldots, a_m$ (referred to as stocks). These items must be delivered to $n$ points $B_1, B_2, \ldots, B_n$ (called consumers), respectively, in quantities $b_1, b_2, \ldots, b_n$ (called needs). It is assumed that the value of transport costs $c_{ij} \geq 0$ $(i = 1, 2, \ldots, m; \ j = 1, 2, \ldots, n)$ for the transport of a unit of cargo (goods) by the supplier $A_i$ is known to the consumer $B_j$. To define the classical transport task, a number of assumptions, detailed in the specialized literature, should be considered, mainly the assumption that there is a balance between stock quantities and needs, i.e.,

$$\sum_{i=1}^{m} a_i = \sum_{j=1}^{n} b_j.$$

For the construction of the economic-mathematical model of the transport problem, variables $x_{ij} (i = 1, 2, \ldots, m; \ j = 1, 2, \ldots, n)$ are introduced. They represent the unknown quantity of product to be transported from the point of departure $A_i (i = 1, 2, \ldots, m)$ to the final point $B_j$ $(j = 1, 2, \ldots, n)$.

The purpose of the classical transport model is to minimize total transportation costs $(Z)$.

The transport task model can be written in the form of final sums as follows:

$$\min Z(x_{ij}) = \sum_{i=1}^{m} \sum_{j=1}^{n} c_{ij} x_{ij}$$

under restrictive conditions:

$$\sum_{j=1}^{n} x_{ij} = a_i (i = 1, 2, \ldots, m)$$

$$\sum_{i=1}^{m} x_{ij} = b_j (j = 1, 2, \ldots, n)$$

$$x_{ij} \geq 0 \ (i = 1, 2, \ldots, m; \ j = 1, 2, \ldots, n).$$

We will not dwell on methods of solving a transportation problem. Appropriate software products are used to determine the optimal solutions in the article.

We will consider the following Example 1. An optimal plan for the distribution of cargo between suppliers and users should be defined, and the following are known:

Three suppliers of one cargo are available, which have respective stocks of 300, 300, and 400 units $(a_1, a_2, a_3)$. Four users are available who have requested needs of 200, 210, 250, and 340 cargo units, respectively $(b_1, b_2, b_3, b_4)$. Between each supplier and user there is a transport link, and the transport costs $c_{ij}$ $(i = 1, 2, 3; j = 1, 2, 3, 4)$ for the carriage of one unit of cargo from the $i$-th supplier to the $j$-th user are given in matrix C:

$$C = \|c_{ij}\| = \begin{Vmatrix} 2 & 6 & 4 & 2 \\ 3 & 1 & 5 & 2 \\ 5 & 4 & 2 & 3 \end{Vmatrix}$$

After entering the variables $x_{ij} \geq 0 (i = 1, 2, 3; j = 1, 2, 3, 4)$ the task model takes the form:

$$min : Z(x_{ij}) = 2x_{11} + 6x_{12} + 4x_{13} + 2x_{14}$$
$$+3x_{21} + x_{22} + 5x_{23} + 2x_{24}$$
$$+5x_{31} + 4x_{32} + 2x_{33} + 3x_{34}$$

under restrictive conditions:

$$x_{11} + x_{12} + x_{13} + x_{14} = 300$$
$$x_{21} + x_{22} + x_{23} + x_{24} = 300$$
$$x_{31} + x_{32} + x_{33} + x_{34} = 400$$
$$x_{11} + x_{21} + x_{31} = 200$$
$$x_{12} + x_{22} + x_{32} = 210$$
$$x_{13} + x_{23} + x_{33} = 250$$
$$x_{14} + x_{24} + x_{34} = 340$$
$$x_{ij} \geq 0 \ (i = 1, 2, 3; j = 1, 2, 3, 4)$$

The instrument Solver in MS Excel is used to determine an optimal solution. The optimal solution for this model is the following:

$$X^1_{opt.} = \|x^1_{ij}\| = \begin{Vmatrix} 200 & 0 & 0 & 100 \\ 0 & 210 & 0 & 90 \\ 0 & 0 & 250 & 150 \end{Vmatrix}, \ Z\left(X^1_{opt.}\right) = 1940.$$

According to this task plan, the first vendor should deliver 200 units to the first user and 100 units to the fourth user; the second supplier must deliver 210 units to the second user and 90 units to the fourth user; and the third supplier must deliver 250 units to the third user and 150 units to the fourth user. Under this transportation plan, the minimum total transportation costs of 1940 are obtained.

The classical model of the transport task assumes that the quantities at suppliers and the quantities at consumers are known. The real situation in business is sometimes different. Let us take, for example, the pharmaceutical business when importing medicines from abroad. All imported medicines come to a national logistics hub. Afterwards, these quantities are distributed to all suppliers. All suppliers move the stocks to consumers (usually corporate customers). This is a classical supply chain in the pharmaceutical business. The classical model of the transport task (with formal description in Example 1) serves partially the decision support process because it assumes known quantities at suppliers and known needed quantities at consumers. The real situation with SCM in pharmaceutical business with a national logistics hub opens the need for extending the classical transport model. Since we have a national logistics hub, we may have different distributions of stock quantities among suppliers. In this case, a different distribution of stock quantities may lead to lower total transportation costs (TTC). To illustrate this idea, Example 2 is created where the stock quantities at suppliers are different from Example 1, but the total quantity of the stock (at suppliers) is the same as in Example 1. The TTC in Example 1 is 1940. The TTC in Example 2 is 1840.

We will consider Example 2, which differs from Example 1 only in the quantities of cargo available at each of the suppliers. Let it be known that each of the three suppliers has 300, 400, and 300 units, respectively.

After entering the variables $x_{ij} \geq 0$ $(i = 1, 2, 3; \ j = 1, 2, 3, 4)$ the task model in this case takes the type:

$$min : Z\left(x_{ij}\right) = 2x_{11} + 6x_{12} + 4x_{13} + 2x_{14}$$
$$+3x_{21} + x_{22} + 5x_{23} + 2x_{24}$$
$$+5x_{31} + 4x_{32} + 2x_{33} + 3x_{34}$$

under restrictive conditions:

$$x_{11} + x_{12} + x_{13} + x_{14} = 300$$
$$x_{21} + x_{22} + x_{23} + x_{24} = 400$$
$$x_{31} + x_{32} + x_{33} + x_{34} = 300$$
$$x_{11} + x_{21} + x_{31} = 200$$
$$x_{12} + x_{22} + x_{32} = 210$$
$$x_{13} + x_{23} + x_{33} = 250$$
$$x_{14} + x_{24} + x_{34} = 340$$
$$x_{ij} \geq 0 \ (i = 1, 2, 3; \ j = 1, 2, 3, 4)$$

The Solver instrument in MS Excel is used again to determine an optimal solution. The optimal solution for this model is the following:

$$X_{opt.}^2 = \left\| x_{ij}^2 \right\| = \left\| \begin{matrix} 200 & 0 & 0 & 100 \\ 0 & 210 & 0 & 190 \\ 0 & 0 & 250 & 50 \end{matrix} \right\|, \ Z\left(X_{opt.}^2\right) = 1840.$$

According to this task plan, the first vendor (supplier) should deliver 200 units to the first user (consumer) and 100 units to the fourth user; the second supplier must deliver 210 units to the second user and 190 units to the fourth user; and the third supplier must deliver 250 units to the third user and 50 units to the fourth user. This transportation plan results in a minimum total transportation cost of 1840.

In the second variant of the task (Example 2), smaller total costs are obtained than those obtained in the first option (Example 1). This shows that the distribution of stocks among suppliers' warehouses results in lowering the total transport costs.

### 3.2. Formal Description of the New Transport Task Model 1

In view of this, in cases where it is permissible to plan in advance the stock of the cargo at the suppliers, we propose the following modified model of the classical transport task: What is specific about this new model 1 is that the quantities available in suppliers' warehouses are unknown. This new model 1 (Figure 1) is applicable when having a national logistics hub in the pharmaceutics business (when importing medicines from abroad) with known quantities in the logistics hub but unknown quantities in suppliers and unknown quantities to be transported (from each supplier to each corporate customer).

The classical transportation task model assumes that the stock quantities at suppliers are known. However, the situation in SCM with pharmaceutical products and a supply chain with a national logistics hub and downstream partners (suppliers and customers) needs a formal description of an extended transportation model. In this work, it is called "new transportation task model 1". Its formal description is as follows:

From $m$ the starting point $A_1, A_2, \ldots, A_m$ (referred to as suppliers), some unknown quantities of homogeneous products must be supplied to several consumers (corporate customers). However, the total sum of these unknown quantities by suppliers is known. This is available at the national logistics hub. These unknown quantities are denoted by $a_1, a_2, \ldots, a_m$ (They are called inventories, stocks, freight, products, goods, and cargo).

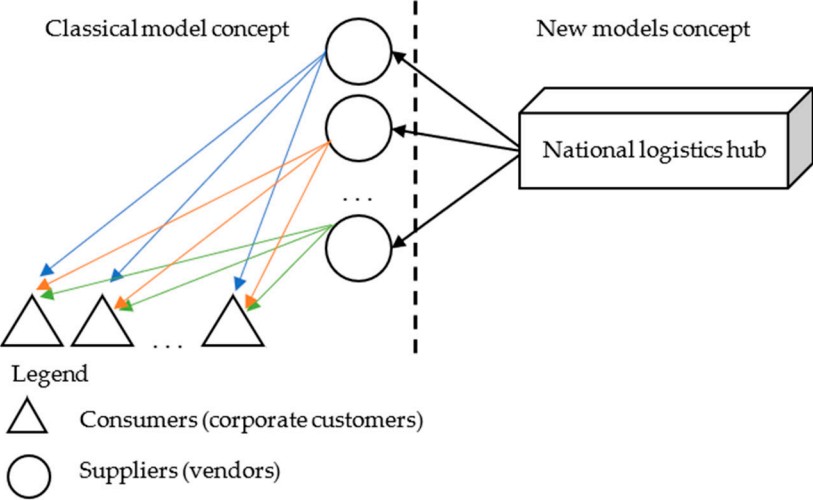

**Figure 1.** The concept of the new two models and the classical transportation model.

One of the assumptions of the classical transport model is that the available quantities at suppliers $(a_1, a_2, \ldots, a_m)$ are known. They are constants in the classical transport model. The new transportation task model 1 assumes that the available quantities at suppliers $(a_1, a_2, \ldots, a_m)$ are unknown. The new transportation task cannot be solved by a classic model because the classical model assumes that the available quantities at suppliers $(a_1, a_2, \ldots, a_m)$ are known, but in actual business practice with a logistics hub (the new transport task), these quantities are unknown. Each transport task has restrictions. The proposed model has all the limitations of the classical transportation task.

There are no restrictions on the quantities that can be provided in advance to each supplier. This product must be delivered to $n$ points $B_1, B_2, \ldots, B_n$ (called consumers, users, and corporate customers), respectively, in quantities $b_1, b_2, \ldots, b_n$ (called needs). It is assumed that the value of transport costs $c_{ij} \geq 0$ $(i = 1, 2, \ldots, m; \ j = 1, 2, \ldots, n)$ for the transport of a unit of cargo (goods) by the supplier $A_i$ is known to the user $B_j$. As in the classical transport problem, a number of assumptions must be considered, mainly the assumption that there is a balance between stocks and needs, i.e.,

$$\sum_{i=1}^{m} a_i = \sum_{j=1}^{n} b_j.$$

For the construction of the mathematical model of the transport problem, variables $x_{ij}(i = 1, 2, \ldots, m; \ j = 1, 2, \ldots, n)$ are introduced, representing the unknown quantity of product to be transported from the point of departure $A_i(i = 1, 2, \ldots, m)$ to the final point $B_j(j = 1, 2, \ldots, n)$. The new transport task model 1 can be written in the form of final sums as follows:

$$\min Z\left(x_{ij}\right) = \sum_{i=1}^{m} \sum_{j=1}^{n} c_{ij} x_{ij} \tag{1}$$

under restrictive conditions:

$$\sum_{j=1}^{n} x_{ij} = a_i \ (i = 1, 2, \ldots, m), \tag{2}$$

$$\sum_{i=1}^{m} x_{ij} = b_j \ (j = 1, 2, \ldots, n), \tag{3}$$

$$\sum_{i=1}^{m} a_i = \sum_{j=1}^{n} b_j, \tag{4}$$

$$a_i \geq 0, \ a_i \leq \sum_{j=1}^{n} b_j \ (i = 1, 2, \ldots, m), \tag{5}$$

$$x_{ij} \geq 0 \ (i = 1, 2, \ldots, m; \ j = 1, 2, \ldots, n). \tag{6}$$

The target function (1) expresses the total transport costs of carrying out the transport of quantities $x_{ij}(i = 1, 2, \ldots, m; \ j = 1, 2, \ldots, n)$ from every supplier to every consumer. The optimal solution is the minimum of the total transport costs (min Z). Restrictive conditions (2) ensure that the transport plan will be adapted to the availability of each of the suppliers, i.e., that the sum of the quantities transported by each supplier to consumers will be equal to its availability. Here we will emphasize once again that the availabilities $a_1, a_2, \ldots, a_m$ are also unknown quantities, but their sum is known—the availability in the national logistics hub. The quantities in the suppliers and the transported quantities from suppliers to consumers should be planned so as to ensure minimal overall transport costs. We assume that there are no restrictions on the capacity of each of the warehouses, i.e., that it is possible to deliver the entire quantity of the product only to a part of the suppliers. In this way, it is possible for some of the suppliers to receive zero amounts of the product. Warehouse capacity restrictions may be for minimal and/or maximum quantities.

Restrictive conditions (3) ensure that the stated needs of each of the users will be met, i.e., that each user will be delivered $b_1, b_2, \ldots, b_n$ units of the product, respectively.

According to the restrictive condition (4), stocks in suppliers will be planned in such a way that the sum of all stocks is equal to the total needs requested by consumers.

The constraint conditions (5) and (6) ensure that the variables in the model are non-negative and that no supplier receives more than the total stated product needs.

The model thus obtained is linear and can again be solved using the methods of linear optimization.

### 3.3. Numeric Example of the New Transport Task Model 1

To show the application of the new transportation model 1 (1)–(6), the following Example 3 is created: We use sample data again from Example 1, but with the condition that the quantities of the product in the warehouses of the suppliers are not known. The total quantity in the national logistics hub is known (1000 units). There are four users who have requested 200, 210, 250, and 340 units of product, respectively. These quantities of the product must be supplied by three suppliers.

Between each supplier and user there is a transport link, and the transport costs $c_{ij}(i = 1, 2, 3; \ j = 1, 2, 3, 4)$ for the carriage of one unit of cargo from the $i$-th supplier to the $j$-th user are given in matrix C:

$$C = \|c_{ij}\| = \begin{Vmatrix} 2 & 6 & 4 & 2 \\ 3 & 1 & 5 & 2 \\ 5 & 4 & 2 & 3 \end{Vmatrix}.$$

It must be determined how many quantities of the product are to be delivered to the suppliers' warehouses so that, after determining a transport plan, a minimum total transport cost is obtained compared to any other option for planning availability with suppliers.

We introduce the variables $a_1 \geq 0$, $a_2 \geq 0$, $a_3 \geq 0$, which express the availabilities of each of the three suppliers. In addition, it is necessary $a_i \leq 1000 \ (i = 1, 2, 3)$, where 1000 is the total availability in the national logistics hub.

After entering the variables $x_{ij} \geq 0 \ (i = 1, 2, 3; \ j = 1, 2, 3, 4)$ expressing the quantities that each supplier must transport to each user, the task model takes the form:

$$min : Z(x_{ij}) = 2x_{11} + 6x_{12} + 4x_{13} + 2x_{14}$$
$$+ 3x_{21} + x_{22} + 5x_{23} + 2x_{24}$$
$$+ 5x_{31} + 4x_{32} + 2x_{33} + 3x_{34}$$

under restrictive conditions:

$$x_{11} + x_{12} + x_{13} + x_{14} = a_1$$
$$x_{21} + x_{22} + x_{23} + x_{24} = a_2$$
$$x_{31} + x_{32} + x_{33} + x_{34} = a_3$$
$$x_{11} + x_{21} + x_{31} = 200$$
$$x_{12} + x_{22} + x_{32} = 210$$
$$x_{13} + x_{23} + x_{33} = 250$$
$$x_{14} + x_{24} + x_{34} = 340$$
$$a_1 + a_2 + a_3 = 1000$$
$$a_1 \geq 0, \ a_2 \geq 0, \ a_3 \geq 0$$
$$x_{ij} \geq 0 \ (i = 1, 2, 3; \ j = 1, 2, 3, 4)$$

The Solver instrument in MS Excel is used to determine an optimal solution. The optimal solution for this new model is the following:

$$a_1 = 200, \ a_2 = 550, \ a_3 = 250$$

$$X^3_{opt.} = \left\| x^3_{ij} \right\| = \begin{Vmatrix} 200 & 0 & 0 & 0 \\ 0 & 210 & 0 & 340 \\ 0 & 0 & 250 & 0 \end{Vmatrix}, \ Z\left(X^3_{opt.}\right) = 1790.$$

From the optimal solution, it can be seen that with the needs of the users (consumers) as stated and the known transport costs, the possible minimum total transport costs for the implementation of the transport plan are 1790. They are obtained when 200 units of the product are provided by the national logistics hub to the first supplier, 550 units of the product are sent from the national logistics hub to the second supplier, and 250 units of the product are sent from the national logistics hub to the third supplier. Following the downstream partners of the supply chain with pharmaceuticals, the following allocations must then be made: the first supplier must deliver 200 units to the first user; the second supplier must deliver 210 units to the second user and 340 units to the fourth user; and the third supplier must deliver 250 units to a third user.

The transport costs of transporting one unit of the product from a specific supplier to each of the users depend on a number of factors, e.g., the distance between the two locations. This means that it is possible that one of the suppliers (labeled $i^1$) has higher transport costs for all users compared to another provider (labeled with $i^2$). Moreover, there will be the following dependence: $c_{i^1 j} \geq c_{i^2 j}$ $(j = 1, 2, \ldots, n)$. It can also be said that a $i^2$ supplier "dominates" a supplier $i^1$, i.e., provides less transport costs to each of the users. In such a situation, it will happen that in the optimal plan of the supplier $i^1$ will be set at zero amount of the product, i.e., $a_{i^1} = 0$. Thus, naturally, the transport task model 1 will show that a supplier must be excluded from the transport plan. It is possible, of course, that other conditions may require each of the suppliers to receive some amount of the product. This can be ensured by imposing additional constraints in transport task model 1.

### 3.4. Formal Description of the New Transport Task Model 2 with Additional Constraints

Depending on the particular conditions of the economic situation itself, other constraints may be added to the restrictive conditions of models (1)–(6). For example, there may be a limit on the amount of product that can be found at each of the suppliers. This condition may be imposed in terms of the supplier's warehouse capacity. Furthermore, a new group of restrictive conditions can be added to model 1. It may be assumed that in each warehouse there may be not less than $a_1^{min}, a_2^{min}, \ldots, a_m^{min}$ units of the product and not more than $a_1^{max}, a_2^{max}, \ldots, a_m^{max}$ units of the product. Furthermore, to model 1 (1)–(6), we add a new group of restrictive conditions (7), and model 2 is created:

$$a_i^{min} \leq a_i \leq a_i^{max} \ (i = 1, 2, \ldots, m). \tag{7}$$

These restrictive conditions ensure that the availability of each of the suppliers will lie within the specified limits between $a_i^{min}$ and $a_i^{max}$ ($i = 1, 2, \ldots, m$).

### 3.5. Numeric Example of the New Transport Task Model 2 with Additional Constraints

To illustrate the new transport task model 2, a new numeric example is created (Example 4).

Let us look at another primer (Example 4). We will use all the sample data from Example 3 by imposing one group of additional restrictive conditions. Let the stocks in the warehouses of each of the suppliers be within the following limits:

$$200 \leq a_1 \leq 400, \ 200 \leq a_2 \leq 400, \ 200 \leq a_3 \leq 400$$

After adding these constraint conditions, the model is again solved using the Solver tool in MS Excel. The optimal solution under these conditions is:

$$a_1 = 400, \ a_2 = 350, \ a_3 = 250$$
$$X_{opt.}^4 = \left\| x_{ij}^4 \right\| = \begin{Vmatrix} 200 & 0 & 0 & 200 \\ 0 & 210 & 0 & 140 \\ 0 & 0 & 250 & 0 \end{Vmatrix}, \ Z\left(X_{opt.}^4\right) = 1790.$$

From the optimal solution thus obtained, it can be seen that, again, the possible minimum total transport costs for the realization of the transport plan are 1790. They are obtained if 400 units of the product are provided as the availability of the first supplier (products sent from the national logistics hub to the first supplier), 350 units of the product as the availability of the second supplier, and 250 units of the product as the availability of the third supplier. Following the supply chain, the following distribution should then be made: the first supplier should deliver 200 units to the first user and 200 units to the fourth user; the second supplier must deliver 210 units to the second user and 140 units to the fourth user; and the third supplier must deliver 250 units to the third user.

We can notice that the minimum total costs of the plan $X_{opt.}^4$ (when there are additional restrictions on the quantities of stocks in the suppliers) coincide with those of the plan $X_{opt.}^3$ (when there are no additional restrictions).

However, this will not always be the case in every business. In general, the imposition of additional restrictions in the model may lead to higher overall costs for the implementation of the transport plan compared to those of the plan without restrictions. For example, if we consider the following additional restrictions (Example 5):

$$200 \leq a_1 \leq 250, \ 200 \leq a_2 \leq 450, \ 200 \leq a_3 \leq 450$$

The optimal solution under these conditions is:

$$a_1 = 250, \ a_2 = 450, \ a_3 = 300$$
$$X_{opt.}^5 = \left\| x_{ij}^5 \right\| = \begin{Vmatrix} 200 & 0 & 0 & 50 \\ 0 & 210 & 0 & 240 \\ 0 & 0 & 250 & 50 \end{Vmatrix}, \ Z\left(X_{opt.}^5\right) = 1840.$$

It can be seen that in this case, the minimum total transport costs for the realization of the transport plan are 1840. They are obtained if 250 units of the product are provided as availability by the first supplier, 450 units of the product as availability by the second supplier, and 300 units of the product as availability by the third supplier. The following allocation must then be made: the first supplier must deliver 200 units to the first user and 50 units to the fourth user; the second supplier must deliver 210 units to the second user and 240 units to the fourth user; and the third supplier must deliver 250 units to the third user and 50 units to the fourth user.

## 4. Results

The paper discusses the known (classical) transport task model, in which the availability of suppliers and the needs of users for a particular product are known in advance. Transport costs from each supplier to each user (consumer) are also known, and an optimal transport plan must be determined on this basis. The classical model of the transport task (with formal description in Example 1) partially serves the decision support process in supply chains with a national logistics hub, several suppliers, and several corporate customers of pharmaceutical products. It assumes known quantities at suppliers and known quantities needed at consumers. The SCM with pharmaceutical products and a national logistics hub open up the need for extending the classical transport model. With a national logistics hub, different distributions of stock quantities among suppliers may exist. In this case, a specific distribution of stock quantities may lead to lower total transportation costs (TTC). To illustrate this idea, Example 2 is created where the stock quantities at suppliers are different from Example 1, but the total quantity of the stock (at suppliers) is the same as in Example 1. In Example 2, the TTC is lower than in Example 1.

On this basis, a new (modified) model 1 of the transport task is constructed, in which a preliminary planning of the available quantities of the transported cargo at the suppliers is made. On the basis of sample data (Example 3), it is shown that through this model it is possible to obtain lower total transport costs for carrying out the transport compared to the classical model of the transport task (Example 1).

The numeric example (Example 3) of the transport task model 1 is designed to check the validity and applicability of the created transport task model 1. The numeric example (Example 4) of the transport task model 2 is designed to check the validity and applicability of the created transport task model 2. Both models have a formal description that allows their adaptation in practice—as part of a software system or as part of a specific calculator in Excel. Both models are adequate for the new SCM reality with a national logistics hub. The obtained results show better corporate logistics in terms of inventory management at the supplier level. The new model 1 assumes that each supplier has an unlimited warehouse. However, in practice, there are limitations. That is why the new model 2 was created. In the new model 2, each supplier has warehouse limitations (minimal and/or maximum quantity). In this case, the newly created model 2 formally describes the real situation in business with importing pharmaceuticals into a national logistics hub and their further distribution in the supply chain with suppliers and customers.

The possibility of adding additional constraint conditions is created by the creation of the new transport task model 2. It describes the existence of constraints in the capacity of suppliers. Two examples (Example 4 and Example 5) are created on the basis of the formal description of the new model 2. In some business cases, additional restrictions in the model may lead to higher overall costs for the implementation of the transport plan compared to those of the plan without restrictions. The proposed two new models have formal descriptions and numerical examples.

The new models do not change the algorithm for calculating the target function (linear optimization). The direct economic effect of the two new models created was acquiring lower total transportation costs compared to the classical model by planning quantities at suppliers and adding restrictions for these quantities. The indirect effect of the two models created is their better formal description of the real business situation in SCM with pharmaceuticals through a national logistics hub and a network of suppliers and corporate consumers.

## 5. Conclusions

Two new (modified) transport task models are proposed in this paper by which lower total transport costs can be provided to perform the transports compared to an application of the classical (baseline) transport task model. The classical model of the transport task assumes that the quantities at suppliers and the quantities at consumers are known. The pharmaceutical business, when importing medicines from abroad into a

national logistics hub, opens up the need for extending the classical transportation model. That is why two new models have been created. The newly created model 1 has known quantities in the logistics hub, unknown quantities in suppliers, and unknown quantities to be transported. The newly created model 2 has known quantities in the logistics hub but unknown quantities at suppliers, unknown quantities to be transported, and restrictions on the quantities at suppliers. The newly created formal descriptions of models 1 and 2 allow their implementation in practice and in ERP systems. The relevant numeric examples prove their efficiency compared to the classical transportation model. Moreover, the two new models better fit the real situation in the pharmaceutical business.

It should not be forgotten that the model of a transport problem, like other mathematical models in economics, is applicable subject to certain conditions and the presence of imposed environmental restrictions. Nevertheless, the application of this model can lead to better economic results in moving material flow from suppliers to consumers. The constructed transport task models can be easily solved using suitable, specially developed software products. Businessmen and practitioners may easily adapt these new models in practice.

The two new models do not consider the costs of transportation between the national logistics hub and the suppliers. Future research may address extending the two new models with a formal description of the costs of transportation between the national logistics hub and the suppliers. In this case, both models will better fit the SCM in the pharmaceutical business.

**Author Contributions:** All authors contributed equally to the paper. All authors have read and agreed to the published version of the manuscript.

**Funding:** This research received no external funding.

**Data Availability Statement:** Not applicable.

**Conflicts of Interest:** The authors declare no conflict of interest.

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
