# Peer review of "Transport Task Models with Variable Supplier Availabilities"

_logistics_

Round 1
Reviewer 1 Report
The article presents very interesting considerations regarding optimizing the supply process in a network system. However, the presented results still require work. I am asking the authors to complete/reflect on the following in their first proofreading:
1. It is good practice for the purpose and structure of the work to be included in the introduction - this is missing in the article.
2. The authors in section 1 state: The novelties of this model are not only theoretical but also practical - in other retail businesses. The novelty in the proposed model gives the possibility for extension of existing models and making them adequate to the up-to-date reality in SCM. In my opinion, the proposed model is very far from practice. I cannot think of a situation where when planning deliveries, I do not know what stock I have at the point of dispatch. Therefore, it is challenging to consider that their use is in line with the current realities of SCM. Please think carefully about this.
3. Section 3 should contain only a description of the method (following the heading), while the results should be presented in section 4 - I understand that the results are presented examples.
4. The discussion is a presentation of arguments that prove the novelty of the presented method, its higher efficiency/effectiveness over other methods, limitations, and barriers to implementation. Section 4, where this discussion was intended to be presented, needs to include these aspects.
5. Also, when preparing the conclusions, I suggest you familiarize yourself with good practices. What are the implementation options? What will be the authors' further research? What has been obtained in the conducted research?
Author Response
Review 1
The article presents very interesting considerations regarding optimizing the supply process in a network system. However, the presented results still require work. I am asking the authors to complete/reflect on the following in their first proofreading:
- It is good practice for the purpose and structure of the work to be included in the introduction - this is missing in the article.
Thank you. The introduction is extended.
- The authors in section 1 state: The novelties of this model are not only theoretical but also practical - in other retail businesses. The novelty in the proposed model gives the possibility for extension of existing models and making them adequate to the up-to-date reality in SCM. In my opinion, the proposed model is very far from practice. I cannot think of a situation where when planning deliveries, I do not know what stock I have at the point of dispatch. Therefore, it is challenging to consider that their use is in line with the current realities of SCM. Please think carefully about this.
In fact, two new models are created. The paper is extended with explanatory text in 3.1 describing the gap (the need) of creating new models with respect to real business situation in retail with pharmaceutical products. A new figure is added and additional explanations in section “3. Methods and research design” for better explanation and clarification of both new models. The models answer the up-to-date reality in SCM of pharmaceuticals with a national logistics hub and downstream partners – suppliers and corporate customers.
- Section 3 should contain only a description of the method (following the heading), while the results should be presented in section 4 - I understand that the results are presented examples.
New texts are added in section “3. Methods and research design” and section “4. Results and discussion”. Some relevant revisions are done.
- The discussion is a presentation of arguments that prove the novelty of the presented method, its higher efficiency/effectiveness over other methods, limitations, and barriers to implementation. Section 4, where this discussion was intended to be presented, needs to include these aspects.
Thanks for the remark. We have added relevant information to prove the better efficiency of the created models.
- Also, when preparing the conclusions, I suggest you familiarize yourself with good practices. What are the implementation options? What will be the authors' further research? What has been obtained in the conducted research?
Thank you. The conclusion section is extended according to the remark.
Reviewer 2 Report
The novelty must be better highlited. The two models must be compared more in detail in the discussion part. The new knowledge introduced by this research must be better pointed.
Author Response
Review 2
The novelty must be better highlighted. The two models must be compared more in detail in the discussion part. The new knowledge introduced by this research must be better pointed.
The paper is extended with explanatory text in 3.1 describing the gap (the need) of creating new models with respect to real business situation in retail with pharmaceutical products. A new figure is added and additional explanations in section “3. Methods and research design” for better explanation and clarification of both new models. The models answer the up-to-date reality in SCM of pharmaceuticals with a national logistics hub and downstream partners – suppliers and corporate customers. New texts are added in section “3. Methods and research design” as well as in section “4. Results and discussion”. Some relevant revisions are done. A new figure is added. More explanations and clarifications are done. The created two transport models are tested, validated and compared to the classical one.
Reviewer 3 Report
The issues of solving transport tasks are always relevant. The study raises a practical question. However, after reading the submitted manuscript, questions arise about the scientific character of the presented material. This is due to the following disadvantages:
1. There is no sufficient justification for the need of this research, i.e., why is this task being solved and for what?
2. Previous studies analysis does not indicate a knowledge gap that should be addressed.
3. What is approach novelty?
4. How does the proposed model in formulas (1-6) differ from the existing one presented in line 127 and below???
5. There is no comparison of results, which is improved, except for entering one constraint.
6. Has the introduction of a new conditional restriction, globally, changed something in the approach or the mathematical formulation of the targeted function?
Also, the manuscript problem is its significant flaw in formal features (see instructions for authors): formulas numbering etc.
Unfortunately, in such a form, the material has no signs of scientific research and must be significantly revised.
Thank you for your invitation
Author Response
Review 3
The issues of solving transport tasks are always relevant. The study raises a practical question. However, after reading the submitted manuscript, questions arise about the scientific character of the presented material. This is due to the following disadvantages:
- There is no sufficient justification for the need of this research, i.e., why is this task being solved and for what?
The paper is extended with explanatory text in 3.1 describing the gap (the need) of creating new models with respect to real business situation in retail with pharmaceutical products. A new figure is added and additional explanations in section “3. Methods and research design” for better explanation and clarification of both new models. The models answer the up-to-date reality in SCM of pharmaceuticals with a national logistics hub and downstream partners – suppliers and corporate customers.
- Previous studies analysis does not indicate a knowledge gap that should be addressed.
The literature review section is extended and precise. The knowledge gap is oriented not only to theory but also to real business. Relevant texts are added; clarifications are done.
- What is approach novelty?
Two new models are created. The answer the gap in theory and practice. Their formal description is extended. Their performance is tested with numeric examples and compared to the classical transport task model.
- How does the proposed model in formulas (1-6) differ from the existing one presented in line 127 and below???
Thank you for the remark. Relevant text is added to make the models clear.
- There is no comparison of results, which is improved, except for entering one constraint.
Comparisons of the results is given in the methods section and results section.
- Has the introduction of a new conditional restriction, globally, changed something in the approach or the mathematical formulation of the targeted function?
Better description of the new model 2 is given.
Also, the manuscript problem is its significant flaw in formal features (see instructions for authors): formulas numbering etc.
Some corrections are done. Thank you.
Unfortunately, in such a form, the material has no signs of scientific research and must be significantly revised.
The whole work is updated precisely with major revisions.
Round 2
Reviewer 1 Report
Section 4 of the revised version discusses the results. It is hard to find elements of discussion of these results there. For this reason, I propose to correct the title of this section and leave only 4. Results.
Author Response
Remark: Section 4 of the revised version discusses the results. It is hard to find elements of discussion of these results there. For this reason, I propose to correct the title of this section and leave only 4. Results.
Response: Thanks for the comment. The title of section 4 is changed. Moreover, the description of the paper structure (at the end of the introduction section) is also changed.
Reviewer 3 Report
Thank you for upgrading the primary draft of the paper. After resubmitting, some improvements are visible. However, the main question is still open. Unfortunately, I was unable to find any changes to the mathematical formalization of a new model. Please note, that if the paper says about a new approach or a new mathematical model, this approach or model should be shown. I appreciate that some numerical examples are provided in the manuscript, but it is not a universal tool that can be used by any researcher.
Please, think about clarity answers to the next questions;
1) What are differs between classic and proposed models in mathematical formulation?
2) Why could not a new transportation task be solved by a classic model? May be good to provide examples of impossible use, or other justifications.
3) Can the proposed model resolve new transportation tasks without limitation, or better to design a restriction system for their application?
4) How will the obtained results by the new model check? What is about model adequacy?
5) What can be written about model validation?
In my opinion, if the answer to the above questions will appear in the manuscript, this study will be more understandable and readable for the scientific community and business.
Check language please
Round 3
Reviewer 3 Report
Some issues were fixed.
Style should be checked.